# Effects of an Innovative Head-Up Tilt Protocol on Blood Pressure and Arterial Stiffness Changes

**DOI:** 10.3390/jcm10061198

**Published:** 2021-03-13

**Authors:** Victor N. Dorogovtsev, Dmitry S. Yankevich, Nandu Goswami

**Affiliations:** 1Federal Research and Clinical Center of Intensive Care Medicine and Rehabilitology, 107031 Moscow, Russia; dorogovtsev.victor3@gmail.com (V.N.D.); yanson_d@mail.ru (D.S.Y.); 2Gravitational Physiology and Medicine Research Unit, Division of Physiology, Otto Loewi Research Center, Medical University of Graz, 3810 Graz, Austria

**Keywords:** pulse wave velocity, hemodynamics, blood pressure, orthostatic intolerance, hydrostatic column

## Abstract

The objective of our study was to identify blood pressure (BP) and pulse wave velocity (PWV) changes during orthostatic loading, using a new the head-up tilt test (HUTT), which incorporates the usage of a standardized hydrostatic column height. Methods: 40 healthy subjects 20–32 years performed HUTT, which was standardized to a height of the hydrostatic column at 133 cm. Exposure time was 10 min in each of 3 positions: horizontal supine 1, HUTT, and horizontal supine 2. The individual tilt up angle made it possible to set the standard value of the hydrostatic column. Hemodynamic parameters were recorded beat to beat using “Task Force Monitor 3040 i”, pulse-wave velocity (PWV) was measured with a sphygmograph–sphygmomanometer VaSera VS1500N. Results: Orthostatic loading caused a significant increase in heart rate (HR) and a decrease in stroke volume (SV) (*p* < 0.05) but no significant reductions in cardiac output, changes in total vascular resistance (TVR), or BP. An analysis of personalized data on systolic blood pressure (SBP) changes in tilt up position as compared to horizontal position (ΔSBP) revealed non-significant changes in this index in 48% of subjects (orthostatic normotension group), in 32% there was a significant decrease in it (orthostatic hypotension group) and in 20% there was a significant increase in it (orthostatic hypertension group). These orthostatic changes were not accompanied by any clinical symptoms and/or syncope. During HUTT, all subjects had in the PWV a significant increase of approximately 27% (*p* < 0.001). Conclusion: The new test protocol involving HUTT standardized to a height of hydrostatic column at 133 cm causes typical hemodynamics responses during orthostatic loading. Individual analysis of the subjects revealed subclinical orthostatic disorders (OSD) in up to 52% of the test persons. During HUTT, all test subjects showed a significant increase in PWV. The new innovative HUTT protocol can be applied in multi-center studies in healthy subjects to detect preclinical forms of orthostatic disorders under standard gravity load conditions.

## 1. Introduction

Practical usage of passive head up tilt test (HUTT) as a tool to investigate “unexplained” syncope began in 1986 [1]. The main purpose of applying such a test was the detection of presyncope or syncope [2,3]. Active orthostatic test from horizontal supine or sitting to standing position has also been used for this purpose [4,5]. Passive orthostatic tests with tilt angles between 60 and 70° were recommended for this purpose. However, they were found to be of low sensitivity and specificity [6]. These issues and limitations have been eliminated via enhancement of orthostatic blood redistribution using different vasodilators: clomipramine [7], isoproterenol [8], nitroglycerine [9], etc. This then led to the development of newer protocols such as Westminster [10], Newcastle [11], Italian [9], etc. These newer protocols emphasize the main purpose of HUTT application: to create more effective conditions for the detection of orthostatic circulatory disorders. Diagnosis of such disorders is based on the differences in systolic blood pressure (SBP) during HUTT and its values in the horizontal position (ΔSBP). Currently, orthostatic hypotension (OH) is diagnosed when ΔSBP reduces by 20 mmHg or more and/or diastolic blood pressure (ΔDBP) by 10 mmHg [12] during standing up (orthostasis). Orthostatic hypertension (OHT) is diagnosed when the ΔSBP is increased by 20 mmHg or more [13]. The choice of such quantitative criteria diagnosis was related to the high probability of clinical manifestations of orthostatic intolerance using this type of protocols.

Orthostatic disorders—OH and OHT are known risk factors for development of cardiovascular diseases (CVD) [14,15]. The most dangerous complication of OH occurs due to acute hypoperfusion of the brain thus leading to dizziness, and even loss of consciousness (syncope). Prevalence of orthostatic hypotension increases from 5% of the population under 50 to 30% or more in persons over 70 years old [16,17]. Currently, tilt table protocols (active and passive) with fixed-angle orthostatic tests to detect orthostatic disorders are widely used. The essence of the traditional protocols was to provoke orthostatic disturbances and to reveal the clinical signs of orthostatic intolerance to confirm an expected diagnosis. Reflex disorders such as vasovagal, situational, and increased sensitivity of carotid sinus are the most frequent causes of the development of orthostatic intolerance and sudden collapse. Many orthostatic disorders also occur because of orthostatic hypotension caused by inadequate arterial hypertension (AH) treatment, hypovolemia of different etiologies, and diseases accompanied by dysfunction of the autonomous nervous system [18]. 

Using traditional “provocative” protocol application, it has been shown that even minor orthostatic disorders in healthy young subjects may be related to the development of the AH [19]. We have developed a special HUTT protocol designed not to provoke orthostatic disorders, but to study orthostatic changes in hemodynamics under standard gravity load conditions. The objective of our study was to identify blood pressure (BP) and pulse wave velocity (PWV) orthostatic changes, using a new head-up tilt test (HUTT), in which the hydrostatic column height was standardized. We consider the results of this study important for the development of preclinical diagnostics of one of the most important risk factors—AH, which is one of the strongest risk factors for development of CVDs [20,21]. 

## 2. Materials and Methods

This study was approved by the Ethics Committee of Federal Research and Clinical Center of Intensive Care Medicine and Rehabilitation, Moscow, Russia, 9 March 2017, Protocol №1/17—2017. All methods in the study were carried out in accordance with the ethical standards expressed in the Declaration of Helsinki of 2013. Written informed consent was obtained from all volunteers prior to study.

Forty young healthy volunteers without arterial hypertension (AH), atherosclerosis or diabetes mellitus performed HUTT experiments (Table 1).

Experiments were carried out in a room with minimal ambient noise. The room temperature was maintained between 24 and 25 °C and the experiments were carried out between 8 and 11:30 a.m. The results of clinical and biochemical blood tests were within normal limits.

In this study, a hydrostatic column height standardized HUTT was applied. The HUTT was performed using an electrically controlled tilt table after 10 min of rest in a horizontal position. The individual tilt angle was determined from the tables for rectangular triangles, in which hypotenuse was the person’s height and the opposing cathetus was the standard hydrostatic column height of 133 cm. After 10 min in HUTT position, the subject returned to the horizontal position 2 for 10 min. 

Hemodynamic parameters were recorded beat to beat throughout the study period: systolic blood pressure (SBP), diastolic blood pressure (DBP), heart rate (HR), stroke volume (SV), cardiac output (CO), total vascular resistance (TVR), using a Task Force^®^ hemodynamic monitor 3040i (CNSystems, Graz, Austria). All data obtained from 6 to 10 min in each of the 3 positions: horizontal l—HUTT—horizontal 2 were averaged. 

In this study we applied OSD diagnostic criteria that were similar in a study conducted on the same age contingent, as those in previous studies [19]. Difference in systolic blood pressure during the test and in horizontal position 1 (ΔSBP) values within ±5 mmHg was taken as the norm—orthostatic normotension (ONT), reduction of this indicator (drop) to −6 mmHg and below was considered as orthostatic hypotension (OH), increase of ΔSBP to +6 and more (rise)—as orthostatic hypertension (OHT). 

In previous studies, in which a standard tilt table protocol was used (e.g., [19]) no measurements of PWV were made. PWV between the heart and an ankle artery was measured with the VaSera 1500 sphygmomanometer at 9–10 min in each of 3 positions.

“Statistica 10” program was used for statistical analyses. The variables that were normally distributed were tested end presented as mean (M) ± standard deviation (SD). Values of *p* < 0.05 were considered as significant.

## 3. Results

### 3.1. Group Analysis of Indicators of Orthostatic Changes in System Hemodynamics and Vascular Stiffness in a Healthy Young Population

All measurement data of the hemodynamics and PWV in the horizontal position 1, at the standardized HUT and in horizontal position 2 are presented in the Table 2.

Differences in selected HUTT parameters as compared to horizontal position 1 were seen (Table 2,)). In HUTT position, compared to the horizontal position, there was a marked increase in HR from 67 ± 11.5 to 72.1 ± 10.4 b/min (*p* < 0.05) a decrease in SV from 93.1 ± 13.9 to 78.6 ± 12.2 mL (*p* < 0.05). Changes in CO and TWR were not significant. Orthostatic changes in healthy volunteers were characterized by minimal changes in blood pressure during HUTT compared to the horizontal position. There was minimal difference between SBP in HUT and horizontal position (ΔSBP) in the group analysis was −1.9 mmHg. The same decrease (−1.9 mmHg) was observed in the DBP. Thus, the group averaged orthostatic changes in hemodynamic parameters corresponded to the ONT criteria.

### 3.2. Personalized ΔSBP Analysis for Evaluating OSD

Considering the importance of the ΔSBP for detecting orthostatic disorders of hemodynamics, we carried out a personalized analysis. In this study, we applied known criteria for such impairments from established protocols [19]. A personalized analysis of ΔSBP revealed various deviations (Figure 1).

The personalized Δ SBP data with the above classification showed no orthostatic disorders only in 19 healthy subjects (48%), ∆SBP corresponded to ONT, in 13 subjects (32%) ∆SBP corresponded to OH and in 8 (20%) to OHT.

In the horizontal position, the PWV was 6.2 ± 0.6 m/s and corresponded to the normal reference values for this age group [20]. At a standardized HUTT position, the PWV increased to 7.9 ± 0.6 m/s (*p* < 0.001). After returning to the horizontal position 2, PWV returned to the initial values of 6.2 ± 0.4 m/s. Comparative analysis of PWV in healthy young subjects with and without orthostatic hemodynamic disturbances in horizontal (6.2 ± 0.7 vs. 6.2 ± 0.4 m/s, *p* > 0.05) and tilt up positions (8.0 ± 1.1 vs. 7.8 ± 0.42, *p* > 0.05) did not show statistical differences.

## 4. Discussion

### 4.1. Key Requirement for Implementing New Protocol 

Discovery of a link between subclinical forms of orthostatic disorders and the development of the AH determines the need for a special HUTT protocol for several reasons. These include:Maximum reduction in the risk of orthostatic disorders with development of clinical signs.Eliminates orthostatic stress related to vertical positions.Sets the standard hydrostatic column height during HUTT for different height persons.Extends the HUTT interval to 10 min so that it includes a period of rapid changes and a period of relative stabilization [21]. The choice of test exposure time was related to the need to orthostatic hemodynamics changes analyze in relative steady state.Applies new criteria for orthostatic disorders adapted to preclinical diagnosis.Includes the mandatory assessment of pulse-wave velocity in monitored parameters during HUTT.

### 4.2. Physiological Background of the New HUTT Protocol

In a horizontal position, the hydrostatic pressure is minimum and only the body thickness of the test subject is determined. During HUTT, it increases in proportion to the tilt up angle and in the vertical position it reaches maximum values. In the vertical position, hydrostatic pressure is minimal at the level of the head and maximum in the distal part of legs. The resulting summation of system and elevated hydrostatic pressure during HUTT results in an increase in venous pressure in the great saphenous vein from 8.7 ± 0.3 mmHg in horizontal position to 47.2 ± 1.0 mmHg during HUTT 30° and 83.9 ± 0.9 mmHg during HUTT 70°. A study has shown that the steady state of venous pressure during HUTT is reached between 64 and 247 s [22]. During orthostasis, arterial blood pressure in the legs also increases by up to 60% [23]. A significant increase in intravascular blood pressure causes a redistribution of blood in the vascular system, with 500–800 mL blood accumulating in the lower body parts [23]. This redistribution of blood reduces the venous return to the heart, the filling pressure in the atria and ventricles of the heart a tendency to decrease systemic BP and activation of cardiopulmonary baroreceptors [24,25]. Orthostatic activation of sympathetic baroreflex causes autonomous nervous system stimulation [26], activity of the sympathetic nervous system (SNS) [27]. Prolonged orthostasis leads to changes in levels of volume regulating hormones, including increases in the renin-angiotensin-aldosterone system (RAAS) hormones [28,29] and vasopressin [29]. Activation of the sympathetic system occurs almost immediately after changes in body position [30], the RAAS is activated after 5–15 min [30]. Orthostatic stimulation of pressor systems is important for increasing peripheral resistance and stabilizing organic blood flow and, above all, of the brain during body positions changes. It is important to emphasize that the degree of orthostatic activation of adaptive systems is directly dependent on the HUTT tilt angle [27].

Thus, in summing up the data presented, the following conclusions can be drawn:Pressor adaptive systems are not activated simultaneously for 0 to 5–15 min after body position changes.Orthostatic stabilization (steady state) of hydrostatic pressure occurs within 5 min after a body position change.The amount of hydrostatic pressure directly depends on the HUTT angle of the test person and reaches its maximum in the vertical position.Degree of orthostatic activation of adaptive neuro-humoral systems is directly dependent on the HUTT tilt angle and exposure time.

The data presented in this study made it possible to clarify the following: The optimal HUTT exposure time, which was 10 min. This exposure is the minimum required to achieve a steady state with orthostatic changes in hydrostatic pressure and the activation of adaptive systems.The need to establish standard values for the height of the hydrostatic column for all test subjects. The application of traditional passive and active orthostatic test protocols does not allow this to be achieved, as differences in test persons height determine differences in the hydrostatic column height.

### 4.3. Orthostatic Changes Cause Transitory Arteries Stiffening

Changes in hemodynamics in young, healthy individuals during new protocol implementation have also been investigated in this study. The group analysis of hemodynamic indicators showed typical changes as have been reported with HUTT application in previous studies. 

Orthostatic activation of pressor systems increases the tone of smooth muscle in vessels, which inevitably leads to increased vascular stiffness. It has been shown that an increase in the angle of HUTT causes a progressive increase in arterial stiffness based on carotid-femoral PWV (cfPWV) measurements [31]. During an active orthostatic test, brachial-ankle PWV (baPWV) increases from 10.9 to 17.6 m/s, by up to 60% [22]. We believe that it is preferable to take measurements of PWV in the baPWV or cardiac-ankle PWV (caPWV) than in cfPWV area when analyzing orthostatic changes. The reason for this choice is that cfPWV gives a measure of stiffness in the elastic artery segment (aorta), which is minimally affected by pressor adaptive systems. At other areas (baPWV and caPWV), measurements are taken both in the aorta and in the arteries of the muscular type. The vasoconstrictor effect of pressor systems in these areas is higher, which results in a greater increase in PWV.

### 4.4. New HUTT Standardization Principle on the Hydrostatic Column Height

Schematically, the method for achieving a standard hydrostatic pressure at HUTT is shown in Figure 2.

Two rectangular triangles of ACD and BCD are conditionally connected by catheters (CD) having one value of hydrostatic column height = 133 cm. The left triangle (Δ ACD) is a schematic representation of the orthostatic inclination of low height persons, hypotenuse of this triangle (AC) is the height of low subjects. The right rectangular triangle (Δ BCD) is for high subjects. The individual tilt angle is determined from the tables for rectangular triangles by the known value of the opposing cathetus equal to 133 cm and by the hypotenuse-height of the subject. An inclination angle X of 60° was set for the lowest test subjects, whose height was 150 cm. The tallest subject should have an inclination angle Y of 42° to obtain the standard height of the hydrostatic column. Using such a protocol (and hydrostatic column) solves two problems: 1. It reduces the risk of developing clinical manifestations of orthostatic disorders by setting the maximum inclination angle of 60° for the shortest subjects and 2. Standardizes the gravitational load by setting a standard value (133 cm) of the hydrostatic column height. Such standardization is crucial.

### 4.5. New Criteria for Orthostatic Disorders in the New HUTT Protocol

In the new protocol, we applied the criteria of orthostatic disorders that are related to the development of the AH over the last eight years [19]. Traditional orthostatic criteria for OH and OHT reflect the marginal degrees of such violations. We believe that the extreme degrees of orthostatic disorders do not develop overnight but develop gradually for many years from slight symptoms to severe ones. The new criteria will make it possible to identify this gradual progression in severity.

### 4.6. Results of the Practical Application of the New HUTT Protocol on of Young Healthy Persons

A study of orthostatic changes in hemodynamics using the new HUTT in young, healthy participants was conducted. The group analysis of data revealed typical orthostatic changes in hemodynamics like the data obtained using HUTT 30–60 and described in the literature [32,33]. During HUTT, there were minor changes in SBP, DBP, reduction in CO, a significant decrease in SV, an increase in HR and an increase in TVR. In the analysis of orthostatic hemodynamics changes, the most important parameter is the ΔSBP. Its value allows us to determine the norm (±5 mmHg), initial signs of orthostatic disorders (+6 or more, and −6 or less mmHg). These criteria made it possible to identify the link between such orthostatic disorders and the development of AH in the eight-year perspective [19], ΔSBP values of ±20 mmHg are risk factors for AH and CVD [34,35]. The results of this study reveal the limitations of the group analysis of data. The average data of the group is normal. A personalized analysis of the ΔSBP parameter found no orthostatic disorders in only 48% of the individuals tested, 20% were identified as having an increased risk of developing AH in the eight-year perspective.

An important part of the new protocol is the measurement of PWV. Transitory increase in PWV during HUTT directly reflects the effect of activation of neurohormone systems; it quickly disappears when the person returns to a horizontal position. During the new HUTT protocol application, there was an average 27% increase in PWV. After HUTT, this indicator returned to its initial level (Horizon 1) 5–10 min after the return to the horizontal 2 position. A person remains in a sitting or standing position for an average of 16 h per day, during this period sympathetic system and RAAS must be activated, which is accompanied by a significant increase in vessel stiffness. Prolonged activation of these systems contributes to the development of AH and CVD [36,37], remodeling the vascular wall and permanent increase in its stiffness [38].

The new HUTT protocol, with new hydrostatic column height standardization and new orthostatic disturbances criteria, was applied for screening young healthy persons. The authors propose to name the new HUTT protocol “Luanda protocol”, the city where the main components of this protocol were developed. The new protocol, as described in Table 3, is designed for preclinical diagnostics of some of the AH and CVD risk factors: preclinical orthostatic disorders and early arterial stiffening. This protocol can be applied in longitudinal and multicenter studies to evaluate orthostatic blood flow regulation using standard gravity load.

## 5. Conclusions

The new test causes typical orthostatic changes in hemodynamics. The personalized analysis revealed subclinical orthostatic disorders in 52% of the test persons. During the HUTT, all the test subjects showed a significant increase in PWV. Further studies of preclinical forms orthostatic dysfunction in healthy subjects are needed, which may include the use of an innovative new protocol in prospective multicenter studies.

## Figures and Tables

**Figure 1 jcm-10-01198-f001:**
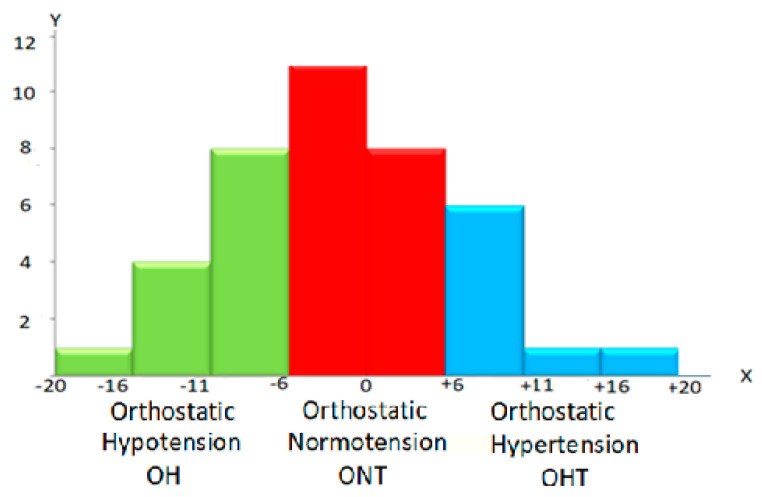
Distribution of subjects by orthostatic changes systolic blood pressure (ΔSBP). *X*—axis—ΔSBP values, *Y*—axis—number of subjects in groups. ONT ±5 mmHg (red columns), OH—6 − 20 mmHg. (green columns), OHT + 6 + 20 mmHg. (blue columns).

**Figure 2 jcm-10-01198-f002:**
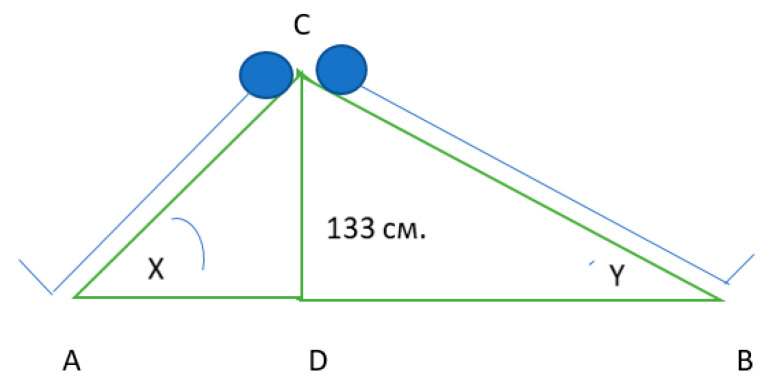
Removal of differences in the height of the test subjects in the Passive orthostatic test (POT), standardized by the height of the hydrostatic column using individual tilt up angles.

**Table 1 jcm-10-01198-t001:** Study population details.

Participants	40
Sex, M/F	23/17
Age, years M ± SD	25.1 ± 2.64
Height, cm M ± SD	173.6 ± 9.93
Body mass index, kg/m2 M ± SD	24.7 ± 3.6

M—mean, ± SD—Standard Deviation.

**Table 2 jcm-10-01198-t002:** Effects of HUTT challenge on selected parameters of the hemodynamics and pulse wave velocity (PWV).

Parameters M ± SD	Position		
	Horizontal 1	Standardized HUTT	Horizontal 2
Measuring Units	M ± SD	M ± SD	M ± SD
SBP mmHg	114.7 ± 11.5	112.8 ± 11.8	116.1 ± 8.5
DBP mmHg	74.8 ± 7.3	72.1 ± 10.4	77.9 ± 6.3
HR b/min	67 ± 11.5	77.3 ± 13.3 *	63.6 ± 10.6
SV ml	93.1 ± 13.9	78.6 ± 12.2 *	95.7 ± 18.7
CO l/min	6.4 ± 1.9	6.0 ± 1.3	6.2 ± 0.9
TVR din·sec·cm^−5^	1078.5 ± 261.9	1168.8 ± 259.0	1092.1 ± 189.0
PWV	6.2 ± 0.6	7.9 ± 0.6 **	6.2 ± 0.4

* *p* < 0.05, ** *p* < 0.001, the head-up tilt test—HUTT, systolic blood pressure (SBP), diastolic blood pressure DBP, heart rate—HR, stroke volume—SV, cardiac output—CO, total vascular resistance—TVR, pulse wave velocity—PWV.

**Table 3 jcm-10-01198-t003:** Comparative characteristics of traditional protocols and the new HUTT protocol. standardized by the height of hydrostatic column.

Comparison Categories	Existing Orthostatic Tests	The New Head-Up Tilt Test Standardized by the Height of Hydrostatic Column
1. Test type	1. Active orthostatic test 2. HUTT standardized by tilt angle	1. Passive orthostatic test standardized by the height of hydrostatic column
2. Purpose of use	Detection of orthostatic disturbances accompanied by clinical symptoms	Detection of subclinical orthostatic disturbances
3. The goal achieving methods	Provocation of orthostatic disturbances accompanied by clinical signs: 1. Active orthostatic test application 2. Increase of HUTT angle up to 80° 3. Increase of HUTT exposition up to 60 min 4. Vasoactive medications application	1. Reducing the risk of developing clinical forms of orthostatic disorders 2.Use of standard height of hydrostatic column = 133 cm 3. Standard HUTT exposition, 10 min 4. Measurement of PWV at all stages of HUTT
4. Diagnostic criteria of orthostatic disturbances	Orthostatic hypotension: drop of SBP by at least 20 mm Hg and/or drop of DBP by at least 10 mm Hg Orthostatic hypertension: rise of SBP by at least 20 mm Hg and/or rise of DBP by at least 10 mm Hg	Orthostatic hypotension: drop of SBP by at least 6 mmHg Orthostatic hypertension: rise of SBP by at least 6 mmHg Orthostatic normotension: Changes of SBP within 5 mmHg
5. Disadvantages	1. Diversity of protocol types and time of exposition. 2. Provide detection of mostly significant orthostatic disturbances	1. Need in additional equipment to measure pulse wave velocity
6. Indications	Type 1: Episodes of syncope in patients without heart conditions Type 2: Diagnosing vasovagal syncope, unexplained falls, vertigo, etc. Type 3: Control of treatment in vasovagal syncope	1. No history of orthostatic disturbances, including syncope 2. Examination of young healthy individuals 2a to detect predictors of arterial hypertension 2b to detect early signs of vascular senescence and risk factors

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
