# Peer review of "Effects of an Innovative Head-Up Tilt Protocol on Blood Pressure and Arterial Stiffness Changes"

_jcm, 2021, doi:10.3390/jcm10061198_

Round 1

Reviewer 1 Report

Dear authors,

your study is of most interest for everybody interested in hemodynamic regulation of arterial wave reflection and central blood pressure regulation. It is now state of good clinical practice and moreover there are so many sex differences known in blood pressure regulation and endothelial function leading to arterial wave reflection as well that you have to present sex disaggregated data. Blood pressure regulation is complex and therefore the paramters have to be measured - it is too complicated to develop scores for prediction of paramters regulated during HUTT for women and men. Otherwise you are not able to draw the right conclusion and give any advice for interpreting HUTT results in practice. 

Please update your results part. Thank you very much. 

Author Response

We wish to thank the reviewer for his enthusiasm and their positive comments regarding the above manuscript. We have now totally re-written the manuscript, and included your comments. We here provide a revised version as suggested by the editor and the reviewers. Our changes are shown in red font, deleted text is shown as deleted and our reply to the reviewers are in bold letters.

  • Blood pressure regulation is complex and therefore the parameters have to be measured - it is too complicated to develop scores for prediction of parameters regulated during HUTT for women and men. Otherwise you are not able to draw the right conclusion and give any advice for interpreting HUTT results in practice.

Done so. In fact, based on your suggestion. We agree with the reviewer that the recommendations and advice on the use of the new test need to be corrected, so the conclusion has been changed:

“Further studies of preclinical forms orthostatic dysfunction in healthy subjects are needed, which may include the use of an innovative new protocol in prospective multicenter studies.”

  • Please update your results part.

 In fact, based on your suggestion, we re-examined the results part and changed it to: In Table 1, the subjects' descriptive values are presented as mean ± standard deviation. In figure 2, the marking error has been corrected. In Table 2, the error of the SV measuring unit has been corrected.

Reviewer 2 Report

  1. Table number 1: sex present only men and in %. are median or media? better to put the significance in subtitles. And it is better to express BMI Ex: Values are n (%) and median (interquartile range). 
  2. Needs a statistical analysis point.   
  3. Table 2: again needs an appropriate scheme and labelling of the parameters, there is no necessary to add another column to put the measurements units. Please check how a table is presented. 
  4. Fig 1: better labelling is needed. 
  5. Conclusions must be reviewed

Author Response

We wish to thank the reviewer for their enthusiasm and their positive comments regarding the above manuscript. We have now totally re-written the manuscript, and included your comments.

  1. Table number 1: sex present only men and in %. are median or media? better to put the significance in subtitles. And it is better to express BMI Ex: Values are n (%) and median (interquartile range).
  2. Needs a statistical analysis point.

Table 1 has been modified as recommended by the reviewer. The subjects' descriptive values are presented as mean ± standard deviation.

  1. Table 2: again needs an appropriate scheme and labelling of the parameters, there is no necessary to add another column to put the measurements units. Please check how a table is presented.

Table 2 has also been modified as recommended by the reviewer.

  1. Fig 1: better labelling is needed.

Figure 1 has been modified as recommended by the reviewer.

  1. Conclusions must be reviewed.

Done so. In fact, based on your suggestion. We agree with your recommendations and advice that the conclusion need to be corrected, so the conclusion has been changed:

“Further studies of preclinical forms orthostatic dysfunction in healthy subjects are needed, which may include the use of an innovative new protocol in prospective multicenter studies.”